

# JAXA Level2 algorithms for EarthCARE mission from single to four sensors: new perspective of cloud, aerosol, radiation and dynamics

Hajime Okamoto[1], Kaori Sato[1], Tomoaki Nishizawa[2], Yoshitaka Jin[2], Takashi Nakajima[3], Minrui Wang[3], Masaki Satoh[4], Kentaroh Suzuki[4], Woosub Roh[4], Akira Yamauchi[4], Hiroaki Horie[5], Yuichi Ohno[5], Yuichiro Hagihara[5], Hiroshi Ishimoto[6], Rei Kudo[6], Takuji Kubota[7], Toshiyuki Tanaka[7]

[1]Research Institute for Applied Mechanics, Kyushu University, Fukuoka, 816-8580, Japan
[2]Earth System Division, National Institute for Environmental Studies, Tsukuba, 305-8506, Japan
[3]Research & Information Center, Tokai University. Kanagawa, 259-1292, Japan
[4]Atmosphere and Ocean Research Institute, The University of Tokyo, Kashiwa, Chiba 277-8564, Japan
[5]Radio Research Institute, National Institute of Information and Communications Technology, Koganei, Tokyo, 184-8795, Japan
[6]Meteorological Research Institute, Japan Meteorological Agency, Tsukuba, 305-0052, Japan
[7]Earth Observation Research Center, Japan Aerospace Exploration Agency, Tsukuba, Ibaraki 305-8505, Japan

*Correspondence to*: Hajime Okamoto (okamoto@riam.kyushu-u.ac.jp)

**Abstract.**

This article gives the overview of the Japan Aerospace Exploration Agency (JAXA) level 2 (L2) Standard and Research algorithms and products by Japanese science teams for EarthCARE Clouds, Aerosols and Radiation Explorer (EarthCARE), which is a JAXA and the European Space Agency (ESA) joint satellite mission. First three single sensor algorithms for 94GHz cloud profiling radar (CPR)-only, 355nm-atmospheric lidar with high spectral resolution function (ATLID)-only, and multi-spectral imager (MSI)-only retrievals, and their products were briefly reviewed. CPR-echo algorithms provide radar reflectivity factor, Doppler velocity, normalized radar cross section and path integral attenuation. CPR-only, CPR-ATLID synergy and CPR-ALTID-MSI synergy algorithms for standard cloud products provide cloud detection, cloud particle type and cloud microphysics, and the research products further provide Doppler velocity, terminal velocity and vertical air motion inside clouds. ATLID standalone algorithms produce aerosol, cloud and clear sky classification products as well as total aerosol extinction and extinction and number concentration of each aerosol types. ATLID-MSI synergy algorithms are developed to retrieve effective radius for each aerosol species in addition to the ATLID-only products. MSI algorithms retrieve cloud effective radius, ice and water content and cloud top pressure. Four sensor algorithms are prepared to produce shortwave and longwave radiative fluxes at the top of atmosphere, those at the surface and also heating rate profiles by using the outputs from CPR, ATLID and their synergy algorithms. The shortwave and longwave fluxes from the four sensor algorithms will then be compared with broad band radiation (BBR) to examine the consistency of the JAXA L2 retrievals.

The algorithms are developed and evaluated by using observational data from satellites and ground-based instruments, and simulation data from the Japanese global cloud-resolving model, the Nonhydrostatic Icosahedral Atmospheric Model (NICAM) with Joint simulator. As for space-borne data, existing space-borne satellites data such as CloudSat, CALIPSO, MODIS and CERES datasets are intensively used. For ground-based observations, High-sensitivity Ground-based Super Polarimetric Ice-crystal Detection and Explication Radar (HG-SPIDER) with a minimum sensitivity of -40dBZ at 15km and over -60dBZ at 1km, Electronic Scanning SPIDER (ES-SPIDER), 355nm high spectral resolution lidar, multiple-field-of-view multiple scattering lidar and Doppler lidars are installed at EarthCARE super-site in Koganei, Tokyo and offers unique opportunities to evaluate and analyse EarthCARE data.



## 1 Introduction

Cloud feedbacks remain to be the largest source of uncertainties in the Coupled Model Intercomparison Projects, across CMIP3 to CMIP6. Dufresne and Bony (2008) performed climate feedback analysis using ensemble of 12 CMIP3/IPCC AR4. Uncertainties of cloud feedbacks, related to low clouds, appeared to be larger than other feedbacks according to their analysis. Both of the multimodal mean and variances in the effective climate sensitivity have increased in CMIP6 compared in CMIP5 (Zelinka et al., 2020). These increases attributed to cloud feedbacks, mainly low clouds. Del Genio (2012) pointed out the parameterizations in GCMs are not sufficient to allow mesoscale convections and the large spread of climate sensitivity attributed to low level clouds does not mean other types of clouds such as high clouds and deep convection do not contribute to the uncertainty in climate sensitivity. Climate model averaged rainfall biases for El Nino events have been reported in IPCC reports across AR3, AR5 and AR6 (Slingo et al. 2022). These seem to show the lack of our understanding of cloud formation processes leading to insufficient parameterizations of convections, low-level water clouds and ice cloud processes. To solve these issues, we need global observing system that measures cloud, aerosol, their radiative properties and vertical velocity to better understand cloud and precipitation formation and their relation to radiation.

Regarding the algorithm developments for JAXA-level2 products, there were several precursors. Prior to the space-borne active sensor observations, ground-based or ship-based observations were used to develop retrieval algorithms for cloud- and aerosol microphysics. Information content of the synergy use of 94-GHz radar and lidar systems was established and the retrieval of ice effective radius and ice water content from radar-lidar synergy was performed for the first time (Okamoto e al. (2000, 2003)) using ground-based 94GHz radar (SPIDER) and Mie-type lidar developed by National Institute of Information and Communications Technology (NICT) and National Institute for Environmental Studies (NIES), respectively. Cloud/hydrometeor mask was developed for 94GHz cloud radar on the vessel Mirai for mid-latitude (Okamoto et al., 2007) and for the Tropical Western Pacific Ocean (Okamoto et al., 2008). Attempts to simultaneously retrieve vertical air motion, terminal velocity and ice cloud microphysics from ground-based Doppler cloud radar–lidar synergy were made, and the estimated vertical air motion has been validated against the Equatorial Atmospheric Radar measurements (Sato et al., 2009). Three-channel aerosol retrieval algorithms for 532nm and 1064nm dual wavelength polarized Mie lidar measurements were developed to uniquely treat the co-existence of aerosol types, i.e., water soluble as small aerosols, sea-salt and dust particles as large particles, and provided the extinction coefficient for each aerosol, and demonstrated using R/V Mirai lidar observations (Nishizawa et al. 2007, 2008).

CloudSat started observations by the first 94GHz cloud radar being in operation since 2006 (Stephens et al. 2018) to study clouds. CALIPSO lidar measures also started measurements with CloudSat and offers attenuated backscattering coefficient 532nm and 1064nm and depolarization ratio at 532nm (Winker et al., 2010). These two spaceborne active sensors provide three dimensional pictures of clouds and aerosols and brought deeper understanding of atmospheric composition and radiative impact of clouds and aerosols (Stephens 2018 and the references therein). The synergetic analysis of CloudSat and CALIPSO also started since the two sensors joined the A-train orbit with small time differences.

Methods developed for ground/ship-based active sensors were applied to CloudSat and CALIPSO observations. A Cloud detection algorithm was developed based on Okamoto et al., (2007, 2008) to create CloudSat and CALIPSO merged data sets (Hagihara et al., 2010, 2014). Vertically resolved phase and ice particle orientations were derived using attenuation and depolarization ratio observations (Yoshida et al., 2010). Cloud particle identification algorithms for CloudSat and CloudSat - CALIPSO synergy was further developed (Kikuchi et al. 2017). Recently, the retrieval of detailed cloud particle shape categories were tested by CALIPSO information (Sato and Okamoto 2023). For cloud microphysics, the radar-lidar synergy algorithms was extended and incorporated functions to deal with specular reflections for CloudSat and CALPSO synergy ice cloud retrievals. The use of CloudSat reflectivity (Ze), backscattering coefficient and depolarization ratio at 532nm from CALIPSO have allowed the investigation of ice cloud microphysics and the ratio of horizontally oriented ice plates to randomly oriented ice crystals in three-dimensional space (Okamoto et al., 2010). The algorithm was further refined to allow the analysis



of cloud regions observed only by CALIOP or CloudSat (Sato and Okamoto 2011). Aerosol typing and microphysical properties retrieval algorithms have been demonstrated from CALIPSO and Aqua/MODIS (Kudo et al., 2023).

EarthCARE Clouds, Aerosols and Radiation Explorer (EarthCARE), the JAXA and the European Space Agency (ESA) joint satellite mission launched in May 20204, carries four sensors, 94GHz cloud profiling radar (CPR), atmospheric lidar (ATLID), multi-spectral imager (MSI) and Broad band radiometer (BBR), on the same platform. The CPR developed by JAXA and NICT uses the same frequency as CloudSat and will measure the Doppler velocity in vertical direction in clouds by pulse-pair method where phase shift of the two successive pulses is measured to infer Doppler velocity. The EarthCARE

CPR will achieve much better sensitivity of -36dBZ than -30dBZ for CloudSat cloud profiling radar due to the low orbit (about 400km) of EarthCARE satellite and large antenna of 2.5m. 355nm-high spectral resolution atmospheric lidar (ATLID) will provide extinction and backscattering coefficient with depolarization ratio. MSI will provide horizontal information of cloud and aerosol information outside of the swarths of CPR and ATLID. BBR provides the radiative fluxes in shortwave (SW) and longwave (LW). Retrieve microphysics of clouds and aerosols will be used as inputs to simulate radiative fluxes in SW and

LW at the top of atmosphere and at the surface. In addition, the heating profiles are also estimated. Comparisons of the estimated radiative fluxes with those actually measured by the BBR offers a unique opportunity to evaluate whether the retrieved quantities about microphysics and radiative characteristics of these particles by CPR, ATLID and MSI.

The EarthCARE observations including Doppler velocity and extinction information inputs will bring new scientific knowledge about the clouds, aerosols, radiation and convections. In the article, we focus on the algorithms used to produce

JAXA Level 2 products by Japanese science teams.

## 2 Description of the algorithms for JAXA L2 products

The structure of products and algorithms used to produce the L2 products and inter relation between products are described in this section. JAXA standard products are produced by relatively mature algorithms and the JAXA research products are produced by advanced algorithms, respectively. Auxiliary products are also prepared and used to produce L2 products. Most

of the products will be processed and reported at a Joint Standard Grids (JSG) in two horizontal and vertical grid spacings. The horizontal and vertical grid spacing of the fine scale version of the products are 1 km and 100 m, respectively. The coarse version of the products is provided after averaging over 10km and 100m horizontal and vertical resolution, respectively. Both of 1km- and 10km-products are reported at 1km. Exception is MSI products reported with 0.5km horizontal grid spacing.


### 2-1 Algorithms for CPR L2a echo products

We first introduce the JAXA CPR level2 algorithm (CPR echo). Both radar reflectivity factor (Ze) and Doppler velocity in CPR L1b product are provided as 500m-integrated values along the satellite path. In order to reduce the random error and improve accuracy of the radar reflective factor and the Doppler velocity, 1-km and 10-km horizontal integration are processed

from CPR L1 products in CPR echo L2 algorithm. Additionally, atmospheric gas attenuation, normalized radar cross section, path integrated attenuation, unfold correction of Doppler velocity, and Doppler correction caused by non-uniform beam filling are also provided in CPR echo L2 products. Center position of the 1-km integration and 10-km integration of the CPR echo L2 products are the same position and those products are provided every 1 km horizontally.

### 2-1-1 Integrated radar reflectivity factor (Ze)

Horizontally integrated received echo power and noise power are calculated using two(1km) and twenty(10km) successive data in L1b products in order to calculate horizontally integrated radar reflectivity factor. Then integrated noise power is subtracted from integrated received echo power and integrated radar reflectivity is derived from the range, the transmit power,



and the radar constant. Finally, the 1-km and 10-km integrated radar reflectivity factor are calculated from the integrated radar

reflectivity and the radar wavelength every 1 km horizontally, and 100 m vertically.

**2-1-2 Integrated Doppler Velocity ($V_D$)**

Similar to Ze, horizontally integrated real and imaginary part of pulse-pair covariance are calculated individually using two(1km) and twenty(10km) successive data in L1b products in order to calculate horizontally integrated Doppler velocity.

Integrated Doppler velocity ($V_D$) is then calculated from integrated real and imaginary part of the pulse-pair covariance as shown in Eq. (1)

$$V_D = \frac{\lambda \cdot PRF}{4\pi} \tan^{-1}\left(\frac{\sum \text{Im}(R(T_s))}{\sum \text{Re}(R(T_s))}\right), (1)$$

where $\lambda$ denotes wavelength of the CPR and is about 3.2mm. PRF is pulse repetition frequency of CPR. Ts denotes time interval between the two pulses. R is the pulse pair covariance corresponding to Ts and is integrated for 500m in L1b. $\sum \text{Im}$

(R) is 1-km or 10-km integrated imaginary part of pulse-pair covariance, and $\sum \text{Re}$ (R) is 1-km or 10-km integrated real part of pulse-pair covariance. PRF is determined by the satellite altitude and the observation window. The satellite altitude changed with latitude, then PRF changed with latitude. For the EarthCARE CPR, we prepared three observation window choices, which is high(-1~20km), mid(-1~18km) and low(-1~16km) mode. PRF can vary between 6100 to 7500Hz. The CPR observation mode can be switched with latitudes automatically. Figure of PRF change with latitude are seen in Hagihara et al., (2023).

Since Doppler accuracy of radar is higher if its PRF is higher for the same Signal to Noise Ratio (SNR) target (Doviak and Zrnic 1993), low mode has higher Doppler accuracy and high mode has lower Doppler accuracy instead.

**2-1-3 Other parameters and possible corrections**

Atmospheric gas attenuation is calculated using temperature, pressure, water vapor profile along with satellite path. In CPR

echo L2 algorithm, vertically integrated atmospheric gas attenuation from top of atmosphere to each height are calculated from meteorological data provided by the ECMWF data, by equation given in the ITUR recommendation P676.

Normalized radar cross section (NRCS) is also estimated in CPR echo L2 algorithm. Surface range position and its reflectivity are estimated by Gaussian fitting of the 1-km horizontally integrated reflectivity profile. From estimated surface range position above, reflectivity of surface range is calculated and it is converted into normalized radar cross section. On the other hand, 10-

km integrated surface position and NRCS are averaged from eleven 1-km integrated surface positions and NRCS.

Measured $V_D$ contains several errors sources such as (1) satellite motion, (2) non-uniform beam filling (NUBF) errors and (3) Doppler folding. Here we focus on (2) and (3). Doppler folding occurs when the velocity exceeds Nyquist velocity ($V_{Nyq}$) determined by $\lambda$ and PRF.

$$V_{Nyq} = \frac{\lambda \cdot PRF}{4}. (2)$$

Since PRF varies as latitude and observation mode changes, $V_{Nyq}$ also shows variation. Therefore, the maximum range of the unfolded Doppler velocity $V_{nyq}$ takes the value from 4.9 to 5.2 m/s for high mode and 5.7 to 6.0m/s for low mode (Hagihara et al., 2023). Due to this, $V_D$ of large precipitation cloud appear as upward velocity while large upward motion could be observed as downward motion, For vertical velocity above $|V_{nyq}|$, unfolding correction is required. Currently simple unfolding

correction scheme is adopted: when $V_D$ of the echo exceeds 3 m/s in upward direction, we use folding correction, i.e. 2*$V_{nyq}$ is subtracted from $V_D$.

Another mechanism for Doppler folding is the NUBF causing $V_D$ to be outside the range of $V_{Nyq}$. If there is a horizontal inhomogeneity of reflectivity factor within certain averaging distance, non-zero vertical Doppler velocity is observed even if there is no actual vertical velocity. The above Doppler error is called NUBF Doppler error. A correction method is developed





using inhomogeneity of Ze information, and the corrected velocity estimated by this method will be provided as the CPR echo research product.

The Path Integrated Attenuation (PIA) denotes the NRCS loss, which is mainly caused by water particle attenuation. We plan to calculate PIA from non-attenuated NRCS map under the orbit. This product will be implemented when the global NRCS data is accumulated adequately.


### 2-2 Algorithms for CPR standalone Cloud products

The standard CPR products include cloud mask, cloud particle type and cloud microphysics. These are derived mainly from radar reflectivity factor ($Z_e$) without using $V_D$. The research products are similar to the standard products but derived by the additional use of $V_D$. The products will be processed in two horizontal and vertical grid spacings. The horizontal

and vertical grid spacing of the fine scale version of the products are 1 km and 100 m, respectively. The coarse version of the products is provided after averaging over 10km and 100m horizontal and vertical resolution, respectively. Both of 1km- and 10km-products are reported at 1km.

### 2-2-1 Algorithms for CPR standalone cloud mask products

The CPR cloud mask algorithms are based on the algorithms originally developed for the analysis of 94GHz cloud radar on the vessel Mirai for the mid-latitude cruise near Japan (Okamoto et al., 2007) and for the Tropical Western Pacific Ocean cruise (Okamoto et al., 2008). The algorithm has been modified to analyse CloudSat 94GHz cloud radar (Hagihara et al., 2010). There are several similarities between our cloud mask algorithm and the method developed in Marchand et al. (2008) which has been used for the hydrometeor detection by CloudSat. EarthCARE CPR cloud mask is extended version of the approach

applied to CloudSat.  The algorithm considers the signal to noise ratio, spatial continuity test (coherent test), and horizontal averaging to identify significant signals from noise and to assign a confidence level. Further details of cloud mask can be found in (Okamoto et al., 2024b).

### 2-2-2 Algorithms for CPR standalone Cloud particle type products

Cloud particle type is conducted by CPR. Details of the algorithm can be found in Kikuchi et al., (2017). Cloud particle type algorithm for CPR uses information of radar Ze from CPR and temperature from the European Center for Medium-range Weather Forecasting (ECMWF) to infer cloud phase, shape and orientation, cloud or water or ice precipitation. The first step used the $Z_e$-temperature diagram for initial hydrometeor type classification. There are eight cloud particle types from CPR; -9: missing, 0:clear, 1: warm water, 2: super cooled water, 3: 3D-ice, 4: 2D-plate, 5: liquid drizzle, 6: mixed phase drizzle, 7:

rain, 8: snow. For example, grid that indicates dBZe>10dBZ and temperature below 0°C is categorized as snow. Further, precipitation correction and spatial continuity test (as coherent filter) are introduced to the hydrometeor type algorithm in order to reduce the misclassification of cloud particle type. Further details of clouds particle type can be found in (Okamoto et al., 2024b).


### 2-2-3 Algorithms for CPR standalone ice cloud category products

After applying the cloud mask and cloud particle type products, the ice particle shape and orientation (there are called ice cloud categories) are inferred by using CPR observables (Sato et al., 2024). The algorithm for CPR only ice cloud category product is trained by using the results from the ATLID-only ice cloud category product where lidar ratio and depolarization

ratio are used to classify the category. The description of ATLID-only ice cloud category is briefly given in 2-2. The retrieved ice particle habit categories are horizontally oriented two-dimensional (2D) plates and their assemblages, 2D columns and



their assemblages, bullet rosettes and three-dimensionally (3D) oriented aggregate types, droxtal/compact types, voronoi/irregular/roughened types, and fractal type snow aggregates.

**2-2-4 Algorithms for CPR standalone cloud microphysics products**

The single scattering properties of ice particles with various shapes and orientations are calculated using the discrete dipole approximation (DDA) for most of the particles (Okamoto 2002, Sato and Okamoto 2006, Okamoto et al., 2010, Sato and Okamoto 2011) and Finite-Difference Time-Domain method (FDTD) for fractal type snow aggregates (Ishimoto et al., 2012) at the CPR wavelength, and multiple scattering effects are estimated based on Sato et al. (2018, 2019).

The ice-phase cloud microphysics retrieval algorithms use forward models for CPR corresponding to the derived ice particle categories. CPR $Z_e$, and $\sigma_0$/PIA are used to determine the ice cloud and precipitation microphysics (Sato et al., 2024). . For each cloudy JSG grid, the cloud microphysics scheme considers the maximum of two different size distributions corresponding to mixed phase, mixture of ice and snow particles, ice and liquid precipitation types, and their co-existence of ice particles with liquid cloud particles. Therefore, in addition to the total effective radius, water content, and number concentrations of the ice and liquid phases, the effective radius, water content, number concentration, and cloud phase of cloud particle with two different size distributions are also reported to capture the complex nature of cloud microphysics.

**2-3 Algorithms for ATLID only products**

ATLID only products include feature mask product, target mask product, Aerosol product, Cloud product, boundary layer height product and aerosol component product. Details of the algorithms can be found in Nishizawa et al., (2024).

There are two horizontal resolutions; 1km and 10km except for feature mask where products with horizontal resolution of 0.3km is also produced in addition to 1km and 10km. In the 10km-resolution product, 10km running-averaged of the 1km observables are performed and the running averaged values are reported at each 1km. Vertical resolution is set to be 0.1km. Prior to deriving the above products, the noise reduction scheme is applied to ATLID L1b data. The noise reduction scheme is a discrete wavelet transform (DWT) type. The DWT has been known to be effective for improving the signal to noise ratio (SNR) of lidar signals (e.g., Fang and Huang 2004).

The following algorithms were developed to produce ATLID L2 products for aerosols and clouds using ATLID L1 data (Nishizawa et al., 2024). The algorithm produces the following five products. (1) Extinction coefficient, backscatter coefficient, depolarization ratio, and lidar ratio of aerosols and clouds at 355nm are estimated by our developed optimization method with Gauss-newton method combined with the line search method (Kudo et al. 2016). (2) Identifiers of layers such as cloud, aerosol or clear-sky layers (Feature mask) are estimated by the combined use of vertically variable criteria method and continuity test method developed for CALIOP analysis (Hagihara et al. 2010; Okamoto et al. 2008). (3) Particle type is classified based on lidar ratio and depolarization diagram for aerosols and signal attenuation (or extinction)-depolarization ratio diagram for clouds (Yoshida et al. 2010, Okamoto et al. 2019). (4) Planetary boundary layer height is estimated using wavelet covariance transform (WCT) method improved for the ATLID analysis. (5) Extinction coefficients of dust, sea-salt, carbonaceous, and water-soluble aerosols at 355nm are retrieved using difference in depolarization and light absorption property of each aerosol component (Nishizawa et al. 2007, 2008, 2011, 2017).

ATLID standalone cloud particle type algorithm enables to create seven cloud particle types from ATLID; -1: missing, 0: clear, 1: warm water, 2: super-cooled water, 3: 3D-ice, 4: 2D-plate, 5: mixture of 2D-plate and 3D-ice, 6: unknown1(water or 3D ice), 7: unknown2. Further details for cloud particle type algorithms can be found in Okamoto et al., (2024b)

Feature mask, Target mask, aerosol, cloud and boundary layer height are categorized as standard products. Extinction coefficient for water-soluble, dust, sea salt and black carbon aerosols are categorized to be research products.

**2-4 MSI standalone product**





There are two types of MSI cloud algorithms, an optimization of the cloud and aerosol unbiased decision intellectual algorithm (CLAUDIA) and the cloud and an optimization of Comprehensive Analysis Program for Cloud Optical Measurements (CAPCOM), to produce MSI standalone standard products. CLAUDIA is used for the discrimination of cloudy and clear sky and CAPCOM is used for the retrieval of cloud optical properties. Aerosol properties are retrieved by Multi-wavelength and multi-pixel method (MWPM) and the product is categorized to be Research ones. When the EarthCARE is in its descending

mode, the nadir pixel of MSI is located around the 102nd pixel counted from the west among 384 pixels, including 48 unusable pixels.

MSI L2a standard products include Cloud Flag and Cloud Phase, optical thickness and effective radius of liquid cloud, optical thickness and effective radius of ice cloud and cloud top temperature. Horizontal resolution is 0.5km. MSI L2a Research products aerosol optical thickness over ocean and land and Angstrom exponent parameter.


### 2-4-1 Algorithms for discrimination of cloudy and clear sky pixels: CLAUDIA

CLAUDIA is originally developed by Ishida and Nakajima (2009). The multiple threshold method is employed for the cloud screening by in CLAUDIA. The CLAUDIA consists of calculation of clear confidence levels for the following three threshold tests, i.e., (1) single reflectance tests, (2) reflectance ratio tests and (3) brightness temperature tests and the integration of these

levels.

For the first single reflectance tests, Reflectance at 0.865μm of the MSI is used over ocean and the reflectance at 0.67μm of the MSI is used over land. Minimum albedo map is constructed from minimum reflectance for a month before the satellite data for cloud screening and the measured reflectance is compared with the minimum albedo, instead of using a static threshold like MOD35 algorithm (Ackerman et a., 1998), in order to reduce the incorrect identification. Lower limit and upper limit

threshold values are determined by using minimum albedo. Reflectance ratio tests use the ratio and the difference of reflectance between two wavelengths. These are effective to detect optically thick clouds. Ratio of reflectance at 0.865μm and 0.67μm and that of 0.865μm and 1.65μm as well as the Normalized Difference of Vegetation Index (NDVI) are considered. Brightness temperature tests rely on the brightness temperature at 10.8μm and are used to detect high clouds over ocean. These tests are applicable to day and night. For all of these three tests, lower limit and upper limit values are set. When the observed value is

larger (smaller) than the upper limit (lower limit), the pixel of interest is clear(cloudy) with high level of clear confidence and clear confidence level is set to be 1 (0). When the observed value is smaller than the upper and larger than the lower limits, the value of clear confidence level is assigned to be between 0 and 1 by linear interpolation. The overall confidence level is estimated by the geometrical mean of all clear confidence levels. Then the cloudy pixels are finally determined according to the value of overall confidence level. When the value is 0, the pixel is considered as cloudy.


### 2-3-2 Algorithms for cloud properties: CAPCOM

CAPCOM is the cloud microphysical retrieval algorithm, originally developed by Nakajima and Nakajima (1995) and Kawamoto et al. (2001). CAPCOM-MSI uses observed radiances of MSI band 1(0.67μm), 3(1.65μm), 4(2.21μm) and 6 (10.8μm) to retrieve cloud optical thickness, water cloud effective radius and cloud top temperature. Ancillary data of vertical

profile of temperature, pressure and water vapor and ground albedo are also used to derive the cloud top height and cloud top pressure. Liquid water path is also calculated by the retrieved cloud optical thickness and effective radius. In order to reduce computing time for the retrievals, look up tables (LUT) are prepared by using one-dimensional radiative transfer code RSTAR (Nakajima and Tanaka 1986) and the retrievals of the above-mentioned cloud properties are carried out by comparing the LUTs and observations.

CAPCOM can be applied to retrieve ice cloud microphysics. Scattering properties of Voronoi ice particles are considered in the retrievals of ice optical thickness and effective radius (Letu et al., 2016, Wang et al., 2023).





The MSI observations has an issue where center wavelength shifts as distortions of spectrum images due to spectral misalignment. This effect is called Spectral MIsaLignmEnt (SMILE). The errors in the retrieved cloud microphysics associated with the SMILE effect were examined by Wang et al. (2023) considering ideal scenes for shallow water and the by using ideal

clouds cases for shallow warm clouds and convective cases. The errors were found to be less than 6%. The synthetic JAXA L1 data were also used with and without SMILE effect for MSI by using NICAM outputs with Joint Simulator for Satellite Sensors (Joint-Simulator) (Hashino et al., 2013, Satoh et al., 2016, Roh et al., 2023) and analyses indicated that the retrieval results are not essentially affected by the SMILE effect.  Joint simulator and NICAM are described in section 2-8.

**2-4 Algorithms for synergy cloud products**

**2-4-1 Algorithms for synergy cloud mask and cloud particle type**

The standard products include Cloud mask, cloud particle type, cloud particle categories and clouds microphysics. The cloud mask results from CPR and those from ATLID were merged to create CPR-ATLID cloud mask. The CPR- and ATLID- type

classification schemes are combined as the CPR-ATLID synergy classification scheme, which is applicable to CPR only detected cloud regions, ATLID only detected cloud regions and CPR and ATLID detected cloud regions. In the CPR-ATLID synergy classification scheme, information of the particle types assigned independently by the CPR- and ATLID- type classification scheme at each observation grid is used to re-classify the particle type based on the developed classification rule. Further, a radar reflectivity and temperature-based melting layer detection scheme is developed and implemented in the

hydrometeor type classification scheme to determine the ice-water mixing layers. Accordingly, the following 14 hydrometeor types are considered from CPR and ATLID synergy: -9: missing, 0: clear, 1: warm water, 2: super-cooled water, 3: 3D-ice, 4: 2D-plate, 5: mixture of 3D-ice and 2D-plate, 6: liquid drizzle, 7: mixed-phase drizzle, 8: rain, 9: snow, 10: water and liquid drizzle, 11: water and rain, 12: mixed phase, 13: unknown, 14: melting layer.

**2-4-2 Algorithms for synergy cloud particle category**

The ice particle habit can be further specified by $\delta$ or by using lidar ratio (S) (i.e.,$S=\sigma_{ext}/\beta$) and $\delta$. We estimated the average backscattering coefficient ($\beta$), backscatter color ratio ($\chi$), and depolarization ratio ($\delta$) for ice particles with a wide range of effective radii for five randomly oriented three-dimensional (3D) and three quasi-horizontally oriented two-dimensional (2D) types of ice particle using physical optics (PO) and geometrical integral equation methods (GOIE) (Okamoto et al., 2019,

2020). The following ice particle habit and orientations (hereafter ice particle categories) are considered; 2D and 3D-Voronoi, Droxtal, 3D-Bullet (and Bullet-Rosette), quasi 2D- and 3D-Columns quasi 2D-Plates, supercooled water and warm water. For CALIPSO, $\delta$-based method has been applied to determine ice particle categories described above (Sato and Okamoto 2023) based on the PO and GOIE calculations. Wavelength dependence of $\delta$ for ice particles in UV and visible wavelengths is relatively small revealed by the PO and GOIE calculations (Okamoto et al., 2020). Thus, similar technique with small

modification of boundaries for each category can be applied to retrieve ice particle categories. The algorithm based on S- $\delta$ relationship at 355nm might produce more reliable classification of ice particle categories (Okamoto et al., 2020).

CPR only particle category algorithm is also developed so that the results of CPR- and ATLID cloud particle categories are combined.

**2-4-3 Synergy algorithms for cloud microphysics**

The retrieval algorithms for cloud microphysics use consistent forward models for each sensor at 94GHz and 355nm corresponding to the derived cloud particle categories. The algorithms are based on the heritage from previously developed methods. ATLID $\sigma_{ext}$, $\delta$, and $\beta$ are used to derive the microphysical properties of  water and ice clouds (Sato et al., 2018, 2019; Sato and Okamoto, 2020), and the synergy among ATLID observables, CPR $Z_e$, and $\sigma_0$/PIA are used to determine cloud and



precipitation microphysics (Sato et al., 2024). As for CPR standalone cloud microphysics products, the effective radius, water content, number concentration, and cloud phase of cloud particle with two different size distributions are also reported in addition to the total effective radius, water content, and number concentrations of the ice and liquid phases.

Cloud microphysics can be further improved by adding the information of MSI. It is also expected to incorporate Doppler velocity information to improve clouds and precipitation microphysics. Simultaneous retrievals of particle fall

velocity and vertical air motion are conducted for CPR-ATLID as demonstrated in Sato et al. (2009) in CPR-ATLID-MSI synergy Research Products.

**2-6 Synergy aerosol products**

ATLID-MSI synergy algorithm was developed to retrieves vertically mean mode-radii for dust and fine-mode aerosols as

well as the extinction coefficients of the abovementioned four aerosol components (Kudo et al., 2023) and will produce the L2 aerosol synergy product. The products are categorized as research one. Products include Extinction coefficients for water-soluble, dust, sea-salt and black carbon as for MSI standard products and mode radii for water-soluble and dust.

**2-7 Four sensor radiative flux products**

Radiative fluxes in shortwave (SW) and longwave (LW) are computed by one-dimensional plane-parallel radiative transfer calculations are performed.  Input data are L2 products of cloud and aerosol profiles from CPR, ATLID and MSI and outputs are radiative flux and heating rate profiles. One dimensional radiative transfer code used in the study is MSTRNX (Sekiguchi and Nakajima 2008, Nakajima et al. 2000). Input parameters include aerosol extinction of four types from ATLID, ice water content, liquid water content, effective radius of ice and water clouds from CPR and ATLID and optical thickness from MSI

as retrieved properties and meteorological conditions, e.g., temperature and pressure, from Auxiliary ECMWF products. The optical parameter table for ice particles assumes Voronoi particles. Details are found in Yamauchi et al., (2024).

**2-8 NICAM/Joint simulator**

The forward simulator called Joint simulator is developed to simulate the JAXA L1 data of signals for all four sensors onboard

EarthCARE (Hashino et al., 2013, 2016, Satoh et al., 2016). It uses the output from NICAM (Tomita and Satoh 2004, Satoh et al., 2008, Satoh et al., 2014). In the simulator, radiative signatures of a vertical column of the atmosphere are calculated with the assumption of a plane-parallel atmosphere. 3D radiative transfer model, MCARaTS can be also chosen (Okata et al., 2016). The basic structure of Joint simulator is taken from SDSU and NASA Goddard SDSU (Masunaga et al., 2010). Concerning the cloud and precipitation properties, a bulk single-moment cloud microphysics scheme with six water categories

(NSW6; Tomita 2008) was applied. Aerosol properties were simulated using the NICAM Spectral Radiation-Transport Model (NICAM-SPRINTARS to simulate global 3D aerosol and their radiative properties (Takemura et al. 2000). Aerosol data include sulfate, soil dust, sea salt, black carbon and organic matter.

Signals of CPR and ATLID are calculated by EarthCARE Active sensor simulator (EASE) (Okamoto et al. 2007, 2008; Nishizawa et al. 2008). MSI signals are given by RSTAR (Nakajima and Tanaka 1986, 1988). BBR signals are by MSTRNX.

Details are described in Roh et al., (2023). The simulator provides synthetic observations for the development and evaluation of the retrieval algorithms for EarthCARE.

**3 Demonstration of the retrieval results and evaluations**

We demonstrate part of the key features of JAXA L2 algorithms to retrieve aerosols, clouds and their radiative properties in

this section.

**3-1 Aerosol properties from ATLID-MSI algorithms**





The ATLID algorithms for aerosols are demonstrated by using CALIPSO and MODIS data obtained in September 22, 2009.
Time height plot of attenuated backscattering coefficient at 532nm, perpendicular component of attenuated backscattering

coefficient at 532nm and attenuate backscattering at 1064nm and results of the target mask is shown in Fig.1. The target mask
algorithm detected thick aerosol layer, high cirrus clouds and low-level clouds. Aerosol component analyses were conducted
using these products (Fig.1). The extinction coefficients of water soluble, light absorbing carbonaceous aerosols, dust and sea-
salt are retrieved by ATLID- MSI algorithms. Co-existence of dust and water soluble is clearly shown. Largest extinction
coefficient was observed due to dust below 5km. Water soluble are also found with smaller extinction.


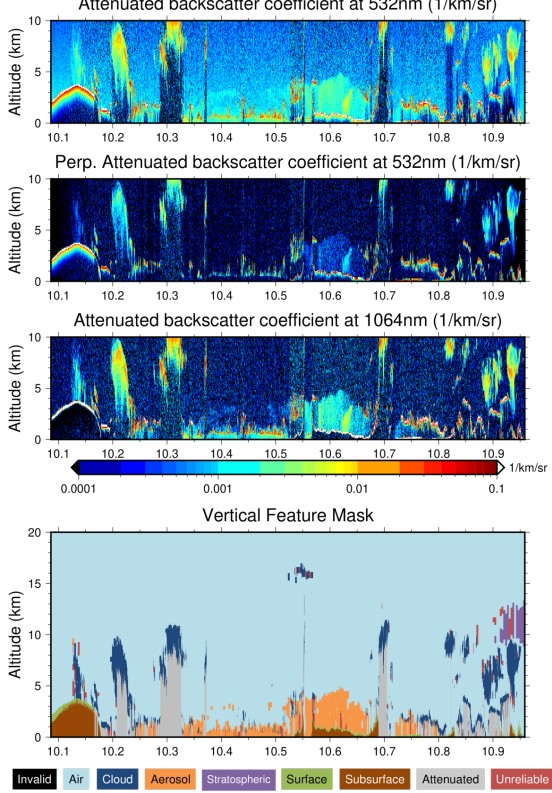

Figure 1. (From top to bottom) CALIPSO observations used to demonstrate EarthCARE ATLID target mask algorithm and
EarthCARE ATLID-MSI aerosol microphysics retrieval algorithms. Time height plot of total attenuated backscattering
coefficient ($\beta_{att}$) at 532nm, (middle) Perpendicular component of $\beta_{att}$ and (bottom) attenuated backscattering coefficient at

1064nm. Result of EarthCARE ATLID Target mask.

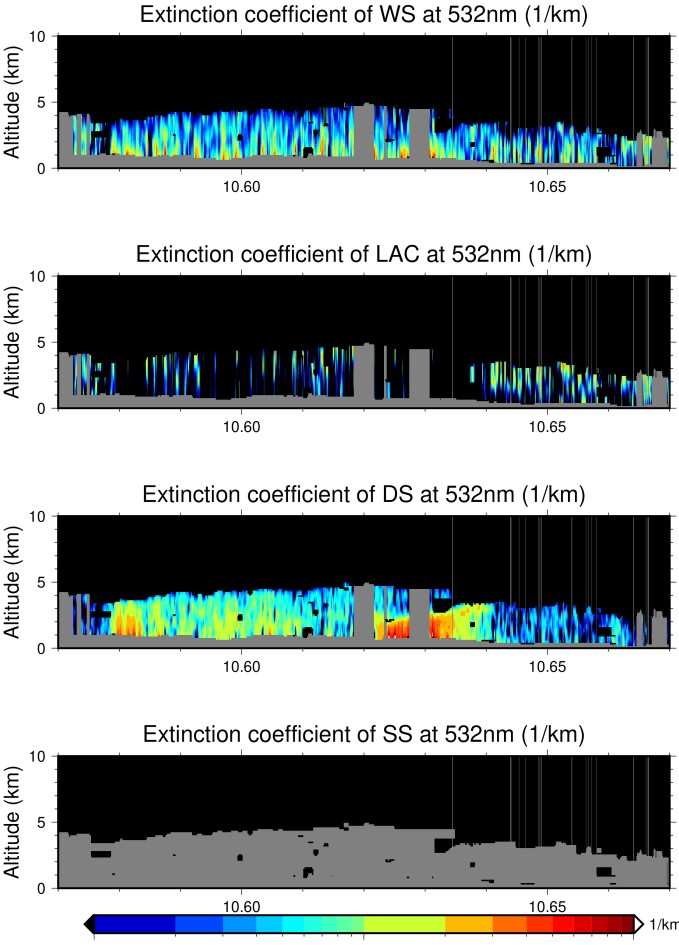

Figure 2. Retrieved Extinction coefficient [1/km] for water soluble, light absorbing carbon, dust and sea-salt (from top to bottom) by EarthCARE ATLID-MSI algorithm for aerosols applied to CALIPSO and MODIS data.

**3-2 Cloud properties from MSI standalone algorithms**

EarthCARE MSI L2 cloud products, which includes horizontal distribution of cloud optical thickness, cloud effective radius, and cloud top temperature, are shown in Fig. 4, Fig. 5, and Fig. 6, respectively. L1 simulated data created by NICAM-Joint simulator on June 19, 2008 are used as input for the retrievals. Both water clouds and ice clouds can be clearly identified in Fig.4, as the clouds from 30 to 33 southern degrees with relatively higher cloud top temperature (260-280K) are water clouds, and the clouds from 33 to 34.5 southern degrees with relatively lower cloud top temperature (220-240K) are ice clouds. Water clouds generally have smaller cloud effective radius than ice clouds, which can be verified in Fig. 5, and in this case the ice clouds are generally shallower than the water clouds, which is shown from Fig. 6.





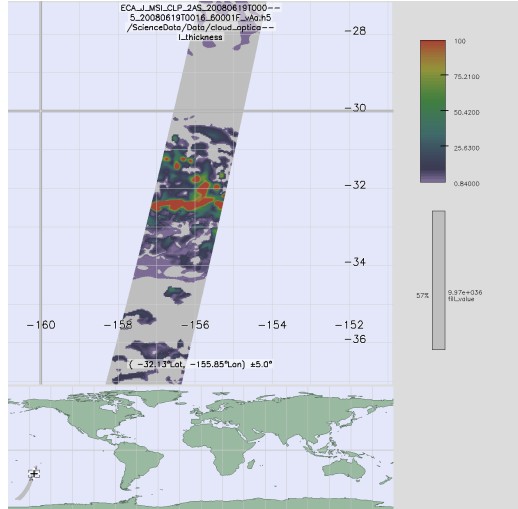

Figure 3. EarthCARE MSI cloud optical thickness product, based on the L1 simulated data by using NICAM-Joint simulator
during 00:00-00:16 UTC, on June 19, 2008.

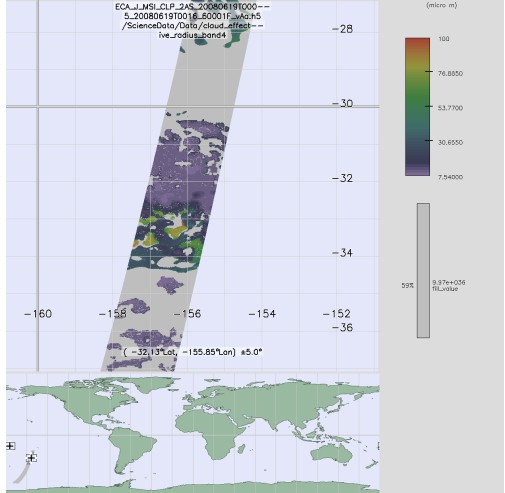

Figure 4. EarthCARE MSI L2 product of cloud effective radius (using MSI band 4), based on the L1 simulated data by using
NICAM-Joint simulator during 00:00-00:16 UTC, on June 19, 2008.






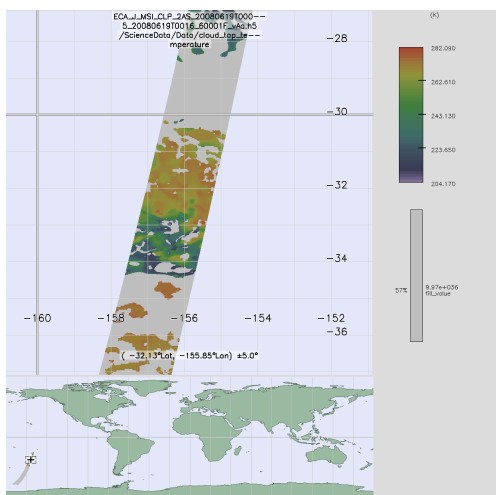

Figure 5. EarthCARE MSI L2 product of cloud top temperature, based on the L1 simulated data by using NICAM/Joint simulator during 00:00-00:16 UTC, on June 19, 2008.

s

### 3-3 Cloud properties from CPR-ATLID algorithms

The EarthCARE CPR-ATLID algorithms were applied to the simulated EarthCARE L1 data from NICAM/Joint simulator and cloud particle type (Fig. 6a), effective radius and cloud water content were derived (Figs. 6b and d). The displayed scene contained ice particles above 4km and rain below 4km. Among NICAM original outputs, only clouds and precipitation scenes

which passed the CPR or ATLID cloud mask were shown as truth for the validation purposes (Figs 6c and e). There was a good agreement between retrieved and simulated values both in ice and precipitation regions.

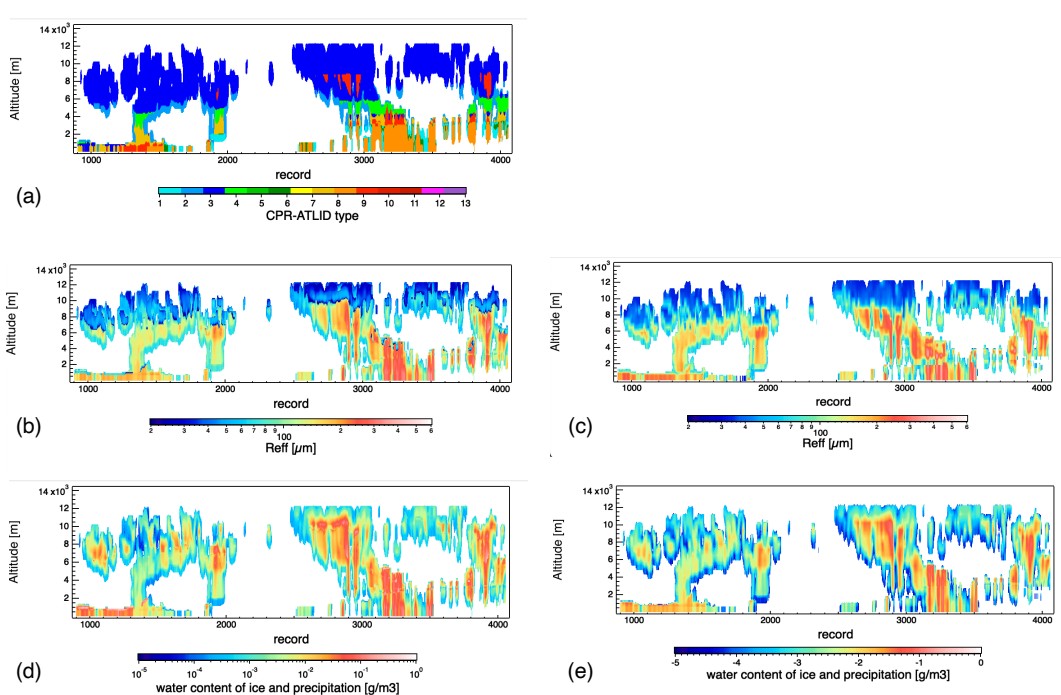





Figure 6 Retrieval results for cloud microphysics from EarthCARE CPR-ATLID algorithms (a) Retrieved cloud particle type,

(b) Retrieved effective radius of ice clouds and precipitation, (c) Effective radius of ice and precipitation particles used to

simulate CPR and ATLID signals by NICAM original outputs after cloud mask, (d) Retrieved ice and precipitation water

content, (e) Ice and precipitation water content used to simulate CPR and ATLID signals by NICAM original outputs after

clouds mask. The simulations are conducted for June 19 2008.


### 3-4 Radiative fluxes and heating rate from EarthCARE four sensor algorithms

 We first show the radiative fluxes and heating rate profiles simulated by the one-dimensional radiative transfer with input of

the EarthCARE L2 products of (1) aerosol microphysics from ATLID-MSI algorithms, (2) cloud microphysics from CPR-

ATLID algorithms and (3) cloud optical thickness from MSI algorithms where L1 simulated data by using NICAM-Joint

simulator are used. There are high clouds and small fraction of low-level water clouds. LW flux is larger than SW flux at the

TOA in this scene (Figure 7).

  Calculation of the radiative fluxes and heating rate profiles are also conducted by using actual observations. That is, input

data are (1) aerosol microphysics from EarthCARE ATLID aerosol algorithms applied to CALIPSO data, (2) cloud

microphysics from EarthCARE CPR-ATLID algorithms applied to CloudSat and CALIPSO data, (3) cloud optical thickness

from EarthCARE MSI algorithms applied to MODIS data, on January 1, 2007. Aerosols are located below 5km and aerosol

extinction coefficient was generally smaller than 0.3. The retrieved aerosol aerosol types are water-soluble aerosols (a mixture

of sulfate, nitrate, and organic aerosols), dust, and sea salt in case of CALIPSO standalone aerosol products. Cloud to height

reaches about 17km in tropics and large positive net radiative heating is found inside the clouds except for cloud top regions

where net radiative cooling is estimated. The results of the SW and LW fluxes at TOA are compared with Clouds and Earth's

Radiant Energy System (CERES) observations. LW fluxes estimated from the retrieved cloud and aerosol microphysics agree

well with that from CERES but there are some discrepancies between simulated and observed SW. These errors might be

reduced by using new retrievals of clouds and aerosols from EarthCARE observations.

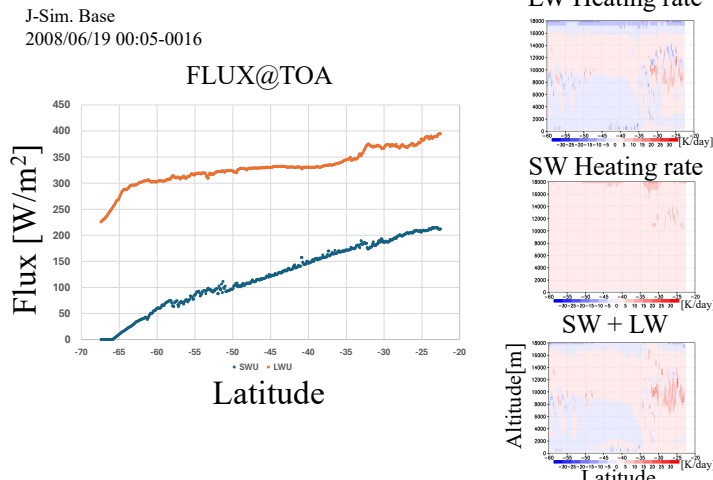


Figure 7. (Left) Upward SW fluxes in blue and upward LW fluxes in red resulting from the RT calculation at the TOA.

(Right top) LW radiative heating (in K/day). (Right middle) SW radiative heating. (Right bottom)  SW+LW radiative heating.



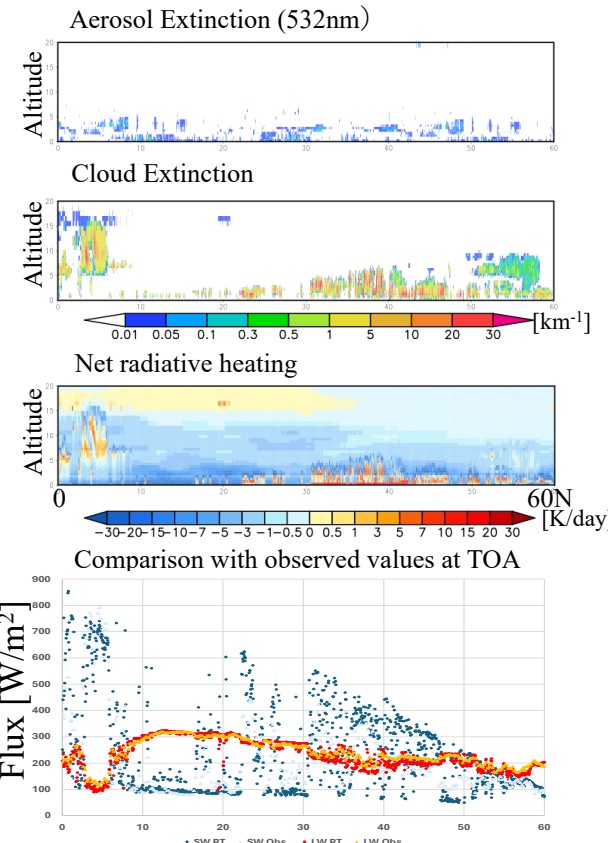

Figure 8.    (From top to bottom) Input data for Aerosol Extinction, Cloud Extinction (in km⁻¹), Net radiative heating (in
K/day) and the SW fluxes in blue and LW fluxes in red resulting from the RT calculation at the TOA. Observed SW fluxes
(light blue) and LW fluxes (light red) from CERES at the TOA are also collocated for comparison.

## 4. Summary

This article gives the overview of the Japan Aerospace Exploration Agency (JAXA) level 2 (L2) algorithms and products by
Japanese science teams for EarthCARE Clouds, Aerosols and Radiation Explorer (EarthCARE), which is a JAXA and the
European Space Agency (ESA) joint satellite mission. The EarthCARE will purvey the first Doppler information by CPR and
together with high spectral polarization function of ATLID, MSI and BBR, the EarthCARE products will contribute to improve
the current climate models, leading to the reduction of the uncertainties in climate predictions and possibly to improve
Numerical Weather Prediction.

The JAXA L2 algorithms produce EarthCARE JAXA L2 products of CPR echo, clouds, aerosols and radiative fluxes/heating
rate as summarized below.

・Radar reflectivity factor and Doppler velocity are derived by algorithms for CPR L2a echo products. Observation of Doppler
velocity is challenging from space and several correction schemes have been developed.



· For the cloud products, one of unique aspects of the algorithms is that depolarization ratio and extinction from ATLID is incorporated into the algorithms to derive detail information of cloud particle shape and orientations. It is expected that these new information will help not only to reduce the retrieval errors in the cloud microphysics but also to understand cloud generation mechanism and improve cloud schemes. CPR standalone, CPR-ATLID and CPR-ATLID-MSI algorithms offer cloud mask, cloud particle type, cloud particle category, cloud microphysics, Doppler velocity, terminal velocity of cloud and

precipitation particles and air motions.

   · ATLID-standalone and ATLID-MSI algorithms offer details of aerosol microphysics. Uniqueness of the algorithms is that these can treat the mixture of several aerosol types including light absorbing carbonaceous aerosols in a same grid and extinction and particle size for each aerosol can be retrieved.

   · MSI standalone algorithms offer water and ice clouds microphysics. Smile effects in MSI measurements are also evaluated

and found to be small.

   · Four sensor algorithms can provide radiative fluxes of SW and LW and heating rate profiles by using the output of JAXA L2 products of microphysics of aerosols by ATLID-MSI and microphysics clouds by CPR, ATLID and MSI.

   The retrievals by these algorithms to create JAXA L2 products are demonstrated by using simulated L1 data from NICAM/J-simulator. Evaluation activities of the L2 algorithms using NICAM/J-simulator outputs were introduced. Further,

A-train data sets are also intensively used to test the algorithms and the four sensor radiative flux products were demonstrated to have good correspondence with CERES observations.

   A comprehensive ground-based synergy observation systems are developed at the super site in NICT in Koganei, Tokyo. The system consists of 94GHz cloud radar, HG-SPIDER, 355nm-Multiple-Field-of-view Multi Scattering Polarization lidar (Okamoto et al., 2016, Nishizawa et al., 2020,) to simulate space-borne lidar signals, 355nm-high-spectral resolution lidar

with polarization (Jin et al., 2021), 355nm-Doppler lidar (Ishii et al., 2022) and 2μm-Doppler lidar (Iwai et al., 2013). There are other data sets, including wind profiler network operated by Japan Meteorological Agency in Japan and K-band radar with disdrometer observations in Antarctica (Bracci et al., 2023) through European Union-Japan collaboration activities. Evaluation of JAXA EarthCARE L2 algorithms using these observations will be reported further.


**Data availability.**

The JAXA EarthCARE synthetic data is distributed from https://doi.org/10.5281/zenodo.7835229 (Roh et al., 2023). The JAXA L2 Echo products, ATLID products, MSI products were processed by the National Institute of Information and Communications Technology (PI: Horie H.), National Institute for Environmental Studies (PI: Nishizawa, T.) and Tokai

University (PI: Nakajima, Y. T.), respectively. CPR, CPR-ATLID and CPR-ATLID-MSI cloud products were processed by Kyushu University (PI: Okamoto, H. O.). The CALIPSO Lidar Level 1B profile data and CloudSat 2B-GEOPROF data used in this study are provided from the NASA Langley Research Center Atmospheric Science Data Center (https://doi.org/10.5067/CALIOP/CALIPSO/LID_L2_05KMCLAY-STANDARD-V4-20) and CloudSat Data Processing Center (https://www.cloudsat.cira.colostate.edu/data-products/2b-geoprof, Marchand & Mace, 2018), respectively. The KU

CloudSat-CALIPSO merged data sets and cloud products, CALIPSO aerosol products, MODIS cloud products are provided and updated to the latest version by JAXA EarthCARE Research A-Train Product Monitor (https://www.eorc.jaxa.jp/EARTHCARE/research_product/ecare_monitor_e.html).

**Author contributions.**

HO coordinated the work and drafted the paper. KS[1] and HO developed the active-sensor based JAXA L2 cloud algorithms. TN[2], YJ, RK processed the L1 data and produced the ATLID and ATLID-MSI L2 products. TN[3] and MW processed the L1 data and produced the MSI L2 products. MS and WR developed the EarthCARE synthetic data and provided the NICAM model outputs. HH, YO and YH processed the L1 data and produced the CPR echo L2a products. KS[4] and AY conducted radiation calculations. HI provided scattering simulations. TK and TT contributed to the validation activities of the algorithms.




**Competing interests.**

The authors have no competing interests to declare.

**Special issue statement.**

This article is part of the special issue "EarthCARE Level 2 algorithms and data products." It is not associated with a conference.

**Acknowledgements.**

The authors would like to thank the JAXA EarthCARE Science Team and the Remote Sensing Technology Center of Japan (RESTEC).


**Financial support.**

This study was supported by The Japan Aerospace Exploration Agency for the EarthCARE mission (grant no. 24RT000246; 24RT000193); JSPS (KAKENHI Grants JP24H00275; JP22K03721); Research Institute for Applied Mechanics, Kyushu University (Fukuoka, Japan).

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
