# Peer review of "JAXA Level2 algorithms for EarthCARE mission from single to four sensors: new perspective of cloud, aerosol, radiation and dynamics"

_Atmospheric Measurement Techniques, 2024_

## Referee Comment (RC3)

As stated in my online comment, the authors provide an overview of the planned products for the EarthCARE mission. Although I understand the need for overview papers such as this for new missions, after review I don't believe that AMT is the correct publication for an article like this. As an overview paper, few details are provided about the nature of the retrieval methods and algorithms. Instead, the reader is most frequently referred to other papers. In the section describing the "synergy" products, in several cases even references to other works are lacking. Very little quantitative assessment of the retrievals is provided. I can't say that the manuscript provides "substantial new concepts, ideas, methods or data" (scientific significance) or results that are "valid" (scientific quality). I think that a descriptive, summary paper like this is probably more appropriate for a different journal (possibly BAMS), but I've recommended "reconsider after major revisions" rather than "reject". My detailed comments follow.

Notes on language: I do find issues with English language usage throughout the paper and they interfere with comprehension. Issues include subject-verb agreement, capitalization, verb tense, grammatical structure, and mistaken or missing articles. Some issues are noted in my comments below, but I did not make note of many of them.

General
+++++++

Some details of the EarthCARE satellite orbit and instrument footprints would be helpful, especially the northernmost and southernmost latitudes reached by the satellite's ground track.

For the "synergy" algorithms, there is a particular lack of adequate detail about the methods.

Section 1
+++++++++

L 18: I think this should be "Earth Cloud, Aerosol, and Radiation Explorer" per the ESA website.

L 22: Usually "path integrated attenuation", I think.

L 34: Is there a citation for a paper describing NICAM?

L 36: Should be "offer" rather than "offers".

L 42: Should be "using an ensemble" rather than "using ensemble".

L 43: Do you mean "larger than uncertainties for other feedbacks"?

L 45: Should be "are attributed". And a citation is needed to define who made this attribution. Was it Zelinka et al.?

L 51: Should be "need a global".

L 69-71: To be accurate, CloudSat and CALIPSO provide only two-dimensional views of clouds and aerosols since both CPR and CALIOP are nadar-pointing instruments.

L 79: Should be "algorithm was extended".

L 85: See comment for L 18.

L 86: Should be "2024".

L 92-93: "horizontal information" and "cloud and aerosol information" - can you use more precise terminology for "information". Also, "swarths" should be "swaths".

L 97-98: "evaluate whether the retrieved quantities..." is not grammatical, please revise.

Section 2:
++++++++++

L 105-108: So, the initial statement is that two vertical grid spacings are used, but then it is stated that both grids use 100 m vertical spacing. What is correct? Also, the 10 km product is reported at 1 km. Does this mean that a horizontal moving-window average is used? Please clarify this.

L 113-115: OK, does this mean that the ground-track footprint of the CPR beam is 500m? How wide is the footprint? Particularly for the 10-km product, how will mixes of cloudy and clear radar bins be handled?

L 146: Is this an ITUR document? Can you provide bibliographic information for it?

L 144-169: Have you made estimates of the residual uncertainties after these corrections have been applied? That would be useful information for data users.

L 152: Why is the correction for satellite motion not being discussed? Given the velocity of the satellite compared to the velocities that are being measured, it seems like satellite motions could be a significant source of error in Doppler velocity.

L 162-164: This is unfamiliar to me. This seems to say that if there is a collection of inhomogenious radar reflectivities within a certain horizontal averaging distance, even if all of those observations have zero Doppler velocity, the Doppler velocity associated with that averaging distance will be non-zero. Is that correct? When you say "averaging distance", are you talking about the 500 m distance over which the imaginary and real parts of the pulse pair covariances are summed (i.e., equation 1)? For that to be true, the sum of the imaginary parts of the pulse pair covariances would have to be zero, I think. Or are you talking about the domain averagine to the 1 km or 10 km horizontal scale? Could you please provide a reference that describes this?

L 172: Can you expand on what is meant by "cloud particle type" and the variables that are included in "cloud microphysics"? OK, I see that these are described more completely in the next several subsections. Maybe add ", described more fully in 2-2-1 through 2-2-4".

L 174-177: The details of the grids have already been presented. It should be sufficient to say the products are produced on the coarse and fine scale grids.

L 205: Do you mean "2-3" instead of "2-2"?

L 210-221: This subsection doesn't really describe the algorithms, only how the scattering properties are determined and the fact that two different size distributions are "considered". Please describe more fully how the retrieval is performed. Also, there is no mention of scattering by liquid cloud, drizzle and rain drops. I would assume Rayleigh and Mie scattering, but this information should be provided.

L 217-219:  Can you clarify this?  What is meant by "considers the maximum of two different size distributions"?  Also how are the size distributions defined?  What are the specific differences of the assumptions used for these different hydrometeor types and mixtures?

L 224-231:  I think most of this paragraph is not needed.  As I noted before, it is not needed to repeat the coarse and fine grid information.   Also, the DWT noise reduction processing is an implementation detail.  Is it already documented in Nishizawa et al. (2004)?  Maybe move the first sentence to the beginning of the third paragraph at L 233.

L 232-233:  So is there one algorithm, or several algorithms?  At the beginning is written "The following algorithms" (plural), then next is written "The algorithm" (singular).

L 239-240:  Is there a reference for the improved WCT method?

L 274-276:  If I understand correctly, this is saying that if a pixel has a reflectance that is larger than the upper reflectance limit, the pixel is classified as clear.  I would expect a pixel with large reflectance to indicate cloud or a snow-covered surface.  Please confirm this.  It might be helpful to describe how the upper and lower limits are determined from the mimimum albedo map.

L 294:  Check this sentence - grammar problems.

L 302-313:  Details of the algorithm are lacking and there is no citation provided to find details.  What are the details of the "developed type classification rule"?  How is the melting layer detection and type classification scheme implemented?

L 324-329:  Perhaps provide a brief description of the retrieval method.  Also, regarding "produce more reliable classification of ice particle categories", what method are you comparing against?  Finally, how are the CPR and ATLID cloud particle categories combined?  Are the ATLID categories used in cloudy layers where the lidar is not attenuated, and the CPR categories used elsewhere, or is something more sophisticated done?

L 330-341:  Please be more explicit.  Is this a combined radar-lidar retrieval?  Or is this a method of combining the results from the distinct radar and lidar retrievals.  If it is a combined retrieval, please provide a description of the method.  If combining results from the distinct retrievals, please provide a description of how that is done.  The details of the CPR standalone retrieval results are not relevant here, I think.

L 344-347:  As before, please provide a succint description of the synergy algorithm retrieval method.

L 350-351:  Grammar problem.

L 356:  I would move the Yamauchi et al., (2024) reference up to the first sentence of this paragraph.

L 359-371:  This paragraph would be clearer by separating the description of the NICAM configuration (i.e., the cloud microphysics scheme, which I assume is not part of the simulator), and the radiative transfer methods used to simulate the sensor observations.

L 383:  Do you mean to refer to Figure 2 here?

L 401:  Should this refer to figures 3, 4 and 5?

L 405-406: How are you evaluating the shallowness of the clouds? The clouds identified as liquid have warmer cloud top temperatures than do the ice clouds, so I would expect that the liquid cloud tops are lower in the atmosphere.

L 400-406: Were these results compared against the NICAM cloud properties? That would seem to be a logical comparison to show here.

L 442: Again, it seems it would have been simple to compare the radiative fields simulated from the EarthCARE L2 products against the same fields from the NICAM simulation. Is there a reason this wasn't done?

L 448: Should be "cloud top height", I believe.

Figure 7: The axis tick labels on the plots need to have a larger font size.

L 468: It's probably more accurate to say that EarthCARE will provide the first Doppler information from a satellite-borne radar, since ground-based Doppler profiling radars are common.

L 474: See my previous comment. Why is there no discussion of corrections for satellite motion?

L 484: "Smile" should be capitalized.

L 492-498: It seems odd to bring up the NICT super site in the final summary of this paper. The super site was only mentioned in the context of providing instrumentation and observations for which precursor retrieval algorithms were developed (L 53-66). This article itself really doesn't focus on the development of the super site.